# Chemical Composition and Biological Activities of *Prangos ferulacea* Essential Oils

**DOI:** 10.3390/molecules27217430

**Published:** 2022-11-01

**Authors:** Natale Badalamenti, Viviana Maresca, Michela Di Napoli, Maurizio Bruno, Adriana Basile, Anna Zanfardino

**Affiliations:** 1Department of Biological, Chemical and Pharmaceutical Sciences and Technologies (STEBICEF), Università degli Studi di Palermo, Viale delle Scienze, ed. 17, 90128 Palermo, Italy; 2Department of Biology, University of Naples Federico II, 80126 Naples, Italy

**Keywords:** Apiaceae, *Prangos ferulacea*, (*Z*)-*β*-ocimene, GC-MS, antioxidant activity, antimicrobial activity

## Abstract

*Prangos ferulacea* (L.) Lindl, which belongs to the Apiaceae family, is a species that mainly grows in the eastern Mediterranean region and in western Asia. It has been largely used in traditional medicine in several countries and it has been shown to possess several interesting biological properties. With the aim to provide new insights into the phytochemistry and pharmacology of this species, the essential oils of flowers and leaves from a local accession that grows in Sicily (Italy) and has not yet been previously studied were investigated. The chemical composition of both oils, obtained by hydrodistillation from the leaves and flowers, was evaluated by GC-MS. This analysis allowed us to identify a new chemotype, characterized by a large amount of (*Z*)*-β-*ocimene. Furthermore, these essential oils have been tested for their possible antimicrobial and antioxidant activity. *P. ferulacea* essential oils exhibit moderate antimicrobial activity; in particular, the flower essential oil is harmful at low and wide spectrum concentrations. They also exhibit good antioxidant activity in vitro and in particular, it has been shown that the essential oils of the flowers and leaves of *P. ferulacea* caused a decrease in ROS and an increase in the activity of superoxide dismutase (SOD), catalase (CAT) and glutathione *S*-transferase (GST) in OZ-stimulated PMNs. Therefore, these essential oils could be considered as promising candidates for pharmaceutical and nutraceutical preparations.

## 1. Introduction

*Prangos* Lindley. is a genus of the Apiaceae family (subfamily of Apioideae), and according to the Plant of the World Online database [1], it comprises forty-eight accepted species that are widely distributed from Portugal to Tibet, with the Irano-Turanian region being the center of diversity [2]. This genus belongs to the *Cachrys* group, which also includes the genera *Cachrys*, *Alolacarpum*, *Bilacunaria*, *Ferulago*, *Diplotaenia*, *Eriocycla*, *Azilia* and *Hippomarathrum* [3,4].

In Iran, Turkey, Iraq and other Asian countries, the genus *Prangos* is used as an aromatic and medicinal herb. Both the roots and the aerial parts of the different species are used and, in the literature, scholars have reported the extensive use of essential oils. These have been used to treat gastrointestinal problems, but their uses as an aphrodisiac, coagulant, carminative and tonic have also been reported [5]. Some articles have described the isolation of non-volatile metabolites, such as coumarins, linear and angular furocoumarins, flavonoids, terpenoids [5], and, very recently, a review on the non-volatile metabolites (coumarins and flavonoids) of *P. ferulacea* (L.) Lindl. (syn. *C. ferulacea* (L.) Calest.) and their biological properties has been published [6].

Regarding volatile oil compositions, the most studied taxon is *P. ferulacea* (L.) Lindl. [6,7], although several other investigations that concern other species have been published [5,8,9,10,11,12,13]. Essential oils have always been widely used for different purposes, not only as ingredients in perfumes or cosmetic applications, but also, and most importantly, for medical purposes. They have shown antibacterial, antifungal, virucidal, antiparasitic and insecticidal properties and function as analgesics, sedatives and anti-inflammatories; therefore, they are widely used in the pharmaceutical industry [14].

*Prangos ferulacea* (L.) Lindl. (synonyms: *Cachrys alata* Hoffm., *C. cylindracea* Guss. ex DC., *C. ferulacea* (L.) Calest., *C. goniocarpa* Boiss., *C. prangoides* Boiss., *Laserpitium ferulaceum* L., *Prangos alata* Grossh., *P. biebersteinii* Karjagin, *P. foeniculacea* C.A. May; *Smyrnium laserpitioides* Crantz) [1] mainly grows in the eastern Mediterranean region and western Asia on arid, stony, mountain pastures, preferentially on basic soils. It is quite widespread in Sicily where it prefers carbonate mountains, above 1000 m of altitude. In Sicily (Madonie Mountains), grazing cattle and sheep eat the species, giving their milk and derived dairy products, such as cheese and salted ricotta, a characteristic smell and flavor [15].

In Turkey, where *P. ferulacea* is known as ‘heliz’, this plant provides a characteristic smell and taste to the very famous cheese ‘Otlu’; furthermore, the addition of this plant provides the cheese with antimicrobial properties [16]. In the central, southern, and eastern parts of Turkey, where its vernacular name is ‘çaşir’, the plant is utilized as a vegetable and is boiled, fried, or pickled [17]. Several etnopharmaceutical properties have been reported in Iran, where *P. ferulacea* (‘jashir’) has been used as a carminative, emollient and tonic for gastrointestinal and liver disorders and has anti-flatulent, sedative, anti-inflammatory, anti-viral, anti-helminthic, antifungal, and antibacterial properties [18]. It is also used in food and yogurt seasoning [19].

In the frame of our ongoing research on the volatiles from the family Apiaceae [20,21,22,23,24], we describe the essential oil composition of an accession of *P. ferulacea* growing in Sicily that has not yet been reported, together with its antioxidant and antimicrobial activities.

## 2. Results and Discussion

### 2.1. Chemical Composition

Fifteen volatile components were identified by GC-MS in the essential oil of flowers from this Sicilian accession of *P. ferulacea*, and they accounted for 96.58% of the total composition (Table 1). The oil was dominated by monoterpene hydrocarbons (13 components, 94.22%), whereas oxygenated monoterpenes (1 component) were represented only by 4-terpineol (2.08%). The amount of sesquiterpene hydrocarbons (one component, 0.28%) reported was negligible. The major constituent that represented about half of the oil was (*Z*)-*β*-ocimene (44.44%). Other monoterpene hydrocarbons that occurred in noteworthy percentages were sabinene (2.80%), 3-thujene (5.79%), and α-pinene (4.28%).

The composition of the oil from the leaves was quite similar. Nineteen metabolites were identified, accounting for 96.22% of the total composition (Table 1). In addition, the essential oil was characterized by a large amount of monoterpene hydrocarbons (13 components, 88.63%), with the largest amount of (*Z*)-*β*-ocimene (61.91%) in comparison to the oil from the flowers. Among the sesquiterpene hydrocarbons (six compounds, 9.59%) the occurrence of caryophyllene should be noted (7.65%), which was completely absent in the oil from the flowers.

In botanical medicine, the presence of *β*-ocimene in essential oils of several plants has been associated with anticonvulsant activity, antifungal activity, and antitumor activity [25,26,27,28]. Ocimene is also a volatile pheromone that is important for the social regulation of honeybee colonies and its involvement in the plant defensive system against herbivore attacks has been reported [29]. Within the Apiaceae family, this component was found to be particularly abundant in the essential oil from *Helosciadium nodiflorum* (L.) W.D.J. Koch (Apiaceae), contributing to the plant’s insecticidal capacity [30,31,32]. 

This analysis showed the occurrence of a new chemotype for this Sicilian accession of *P. ferulacea* that is characterized by a large amount of (*Z*)-*β*-ocimene. A previous investigation into another Sicilian accession showed the presence of two isomers of *β*-ocimene [7], whereas Seidi Damyeh et al. [33], using an essential oil sample from Iran, reported only the *E*-isomer of *β*-ocimene as the most abundant volatile component (28.3%). Considerable variability in the chemical profiles of the essential oils of the populations of *P. ferulacea* of different geographical origin has been reported and recently reviewed [6]. In fact, for a Turkish accession of *P. ferulacea*, 2,3,6-trimethyl benzaldehyde was reported as the predominant constituent (66.59%) [34], whereas *β*-pinene (43.1%), α-pinene (22.1%) and *δ*-3-carene (16.9%), (*E*)-caryophyllene (48.2%), α-humulene (10.3%) and spathulenol (9.4%), terpinolene (38.1–56.3%), α-pinene (57%), and α-pinene (36.6%) and *β*-pinene (31.1%), respectively were detected as volatile markers of different Iranian populations [6]. The chemical diversity between the different oils compared is linked to many factors, such as pedoclimatic conditions, including the climate, amount of rainfall, altitude, distance and effects of proximity to the sea and exposure to wind and sun. Sabinene, the second most abundant component in our oils, was detected among the main constituents only in the essential oil of the fruits of a Sardinian population (15.9%) [35]. 

### 2.2. Essential Oil Antimicrobial Activity 

Some studies show that plants extracts or components that belong to the genus *Prangos* exert significant antibacterial, antifungal, antioxidant, anti-inflammatory, hypoglycemic and analgesic activities [5,6]. This study evaluates the antimicrobial and antioxidant activity of *P. ferulacea* essential oils from its flowers and leaves. The essential oils were tested using vital cell counts to produce data for dose–response curves and the MIC assay. In the first experiment, *P. ferulacea* oils were used against the following Gram-positive strains: *Staphylococcus aureus* ATCC6538P, *Bacillus subtilis* AZ54, *Bacillus cereus* ATCC10987, and Gram-negative strains, such as *Escherichia coli* DH5α, *Pseudomonas aeruginosa* PAOI, and *Salmonella tiphymurium* ATCC14028. We carried out experiments using the vital count method to create the dose–response curves shown in Figure 1. This figure shows that the oil extract from the flowers (panel A) and leaves (panel B) exhibits dose-dependent antimicrobial activity, the oil extracted from the flowers exhibits good antimicrobial activity against both Gram-positive and negative strains, while the oil extracted from the leaves exhibits greater antimicrobial activity directed against Gram-positive strains. In general, the most sensitive bacteria are the Gram-positive bacilli, which are almost completely killed at a concentration of 200 µg/mL with both essential oils. More details can be found in Appendix A.

The antimicrobial activity of the *P. ferulacea* oils was also analyzed according to the broth microdilution method. The minimum inhibitory concentration (MIC) values of the oil from *P. ferulacea* flowers were found to be between 100 and 200 µg/mL for Gram-positive and Gram-negative bacteria, as shown in Table 2. For the *P. ferulacea* essential oil extract from its leaves, MIC values of 100 and 200 µg/mL were reported against Gram-positive and Gram-negative bacteria. The lowest oil concentrations that inhibited bacterial growth were recorded for the Gram-positive bacilli, according to our previous experiments. Therefore, essential oils that are abundant in terpenes have been shown to possess remarkable antimicrobial activity. However, the weak activity against Gram-negative bacteria could be due to the presence of their outer and inner membranes that protect the Gram-negative bacteria from the effect of the oil components [36].

Different studies have reported that extracts, essential oils and pure compounds of the *Prangos* species have shown strong antibacterial, antifungal, and antiviral activity. For example, Gram-positive bacteria such as *S. aureus* and *B. cereus* were inhibited by various *Prangos* species, such as *P. pabularia* and *P. platychlaena*. In a study performed with different extracts of *P. hulusii* roots collected in Turkey, the strongest antibacterial activity was associated with the dichloromethane extract of the plant with *E. coli* with an MIC value of 0.156 mg/mL [37,38].

Regarding the composition of the oils used in our study, monoterpene hydrocarbons accounted for 94.17% and 88.63% for the flowers and leaves, respectively, which could be responsible for most of the antimicrobial activity. However, when the sample under analysis is a complex mixture, it is difficult to attribute the activity to a single molecule or to a group of constituents. Furthermore, in the literature, there is evidence that minor components with synergic action play a critical part in antimicrobial activity [39]. So, for this reason, it is likely that the bioactivity of our oils is due to a synergistic effect between the different compounds present in *P. ferulacea* essential oil [5].

### 2.3. Essential Oil Antioxidant Activity 

Free radicals produced in all organisms can cause oxidative damage to biological molecules such as DNA, fatty acids, and amino acids. Consequently, oxidative stress plays an important role in the development of some chronic diseases [39], so plants or their extracts with antioxidant activity can play a role in the protection of the health of living organisms. Due to the presence of hydrocarbon monoterpenes in the *P. ferulacea* essential oils, its radical scavenging ability was evaluated. Figure 2 shows the increasing the % of scavenging activities of ABTS and H_2_O_2_ radicals as the concentration of oil (1–1000 µg/mL) increases.

The data shown in Figure 2 are expressed in Table 3 as IC_50_ values, which demonstrate the oil concentration that causes a 50% reduction in ABTS and H_2_O_2_ radicals. The *P. ferulacea* essential oil extract from flowers shows anti-H_2_O_2_ activity with IC_50_ values of 60 µg/mL and the highest anti-radical effect (IC_50_ value of 100 µg/mL) for ABTS. *P. ferulacea* leaf oil shows anti-H_2_O_2_ activity with IC_50_ values of 50 µg/mL and the highest anti-radical effect (IC_50_ value of 500 µg/mL) for ABTS. 

### 2.4. Antioxidant Enzymes Measured in PMN Cells

The antioxidant activity of *P. ferulacea* was investigated by testing the essential oils from flowers and leaves on OZ-stimulated PMNs to induce oxidative stress. Both ROS levels and the activity of SOD, CAT and GST enzymes were evaluated. As can be observed from Figure 3, as a result of the stress induced by OZ, there is a significant increase in ROS, but by treating the PMN with the flowers’ oil, a gradual reduction that is proportional to the increase in concentration is observed. In fact, in the PMNs treated with 25 µg of flower essential oil, a significant reduction in ROS levels can also be observed, and in addition, in the PMNs treated with 100 µg of flower essential oil, ROS levels comparable to the control (non-stressed PMN) can be observed. The leaf oil, on the other hand, induces a significant reduction in ROS levels to 100 µg.

As for the activity of antioxidant enzymes in PMNs treated with essential oils extracted from flowers, they show the same trend. In fact, as can be observed from Figure 4, the activity of CAT SOD and GST statistically increases as the concentration of flower essential oil increases. As for the essential oil of leaves, the activity of SOD and CAT increases in the PMNs treated with 50 and 100 µg compared to the PMNs stimulated with OZ, which were not treated with essential oils. With regard to the activity of GST, it increases in PMNs treated with 50 µg of leaf essential oil compared to PMN that is stimulated with OZ but not treated with essential oils.

Although the main components are expected to be mainly responsible for the antioxidant activity of an essential oil, the key role of some minor components in decreasing or increasing the antioxidant activity cannot be ignored [40].

Ruberto and Baratta [41] have studied the antioxidant efficacy of one hundred pure components of essential oils. The main classes of compounds were analyzed, namely monoterpene hydrocarbons, oxygenated monoterpenes, sesquiterpene hydrocarbons, oxygenated sesquiterpenes, benzene derivatives and *nor*-isoprenoid components, including alcohols, aldehydes and ketones, which represent the most common constituents of essential oils.

In the work of Çoruh et al. [42], the antioxidant capacities of *Heracleum persicum* Desf., *Prangos ferulacea* (L.) Lindl. and *Chaerophyllum macropodum* Boiss were evaluated. From this comparison, it emerged that *P. ferulacea* was a better antioxidant than the other two plants in the family. The results of our study showed high antioxidant activity, expressed as the activity of the SOD, CAT, and GST enzymes on the PMNs and, in particular, significant variation in the antioxidant capacity of the different parts of *P. ferulacea* emerged. The results revealed that the organ can significantly affect their antioxidant abilities. The highest antioxidant activity was recorded for the essential oils of flowers, which can be considered natural antioxidants with higher activity than the essential oils of leaves.

It is assumed that the difference in antioxidant capacity between the two organs of the plant is due to the different compositions of essential oils. The antioxidant efficacy of *γ*-terpinene, camphene and *β*-ocimene present in some essential oils has already been demonstrated by Negi et al. in 2012 [43]. α-pinene and limonene may also be responsible for the antioxidant potential [44]. This would explain why greater activity was demonstrated by the extracts of essential oils of flowers compared to those of leaves. In fact, the compounds mentioned are present in greater quantities in the essential oils of the flowers compared to the essential oils of the leaves, with the sole exception of *β*-ocimene, which is also present in high quantities in the essential oils of the flowers.

## 3. Materials and Methods

### 3.1. Plant Material

Flowers (470 g) and leaves (250 g) from twenty individuals of *Prangos ferulacea* (L.) Lindl., covering about 200 m^2^, were collected at Piano Zucchi, Palermo, Sicily, Italy, at about 1100 m s/l, 37°53′51″ N; 13°59′50″ E, in June 2022. The samples, identified by Prof. Vincenzo Ilardi (Department STEBICEF, University of Palermo, Italy) by comparisons with the descriptions reported in books and authentic samples, have been stored in the Herbarium of the University of Palermo (Voucher No. PAL 109762).

### 3.2. Isolation of Volatile Components

Extraction of essential oils was carried out according to the method of Basile et al. [45]. Using a Waring blender, the dried samples, immersed in 0.5 L of distilled water, were extracted by hydrodistillation following the methods reported in [46]. The dried oils were sealed in vials and stored in the freezer (−20 °C) until the time of analysis. The yields of the two oils were 0.62% and 0.28% (*w/w*) for flowers and leaves, respectively.

### 3.3. GC-MS Analysis 

Analyses of essential oils were performed according to the procedure reported by Badalamenti et al. [47]. 

### 3.4. Bacterial Strains

The Gram-negative strains *Escherichia coli* DH5α, *Pseudomonas aeruginosa* PAOI and *Salmonella tiphymurium* ATCC14028 and Gram-positive strains *Staphylococcus aureus* ATCC6538P, *Bacillus subtilis* AZ59 and *Bacillus cereus* ATCC10987 were used to evaluate antimicrobial activity. These strains were grown in LB medium, under agitation at 37 °C. 

### 3.5. Antimicrobial Activity Assay

The method used to evaluate the antimicrobial activity was the cell viability counting method of bacteria [48]. Gram-positive and Gram-negative strains were incubated with both essential oils at different concentrations (1, 10, 100, and 200 µg/mL). Bacterial cells without essential oils represented the positive control and cells with DMSO at 80% concentrations were used as the negative control. Each experiment was carried out in triplicate and the reported result was an average of three independent experiments. (*p* value was <0.05).

### 3.6. Determination of Minimal Inhibitory Concentration

The microdilution method established by the Clinical and Laboratory Standards Institute (CLSI) was used to evaluate of minimal inhibitory concentrations (MICs) of *P. ferulacea* essential oil against the Gram-positive and Gram-negative strains. In addition, ~5 × 10^5^ CFU/mL was added to 95 µL of Mueller–Hinton broth (CAM-HB; Difco) with or without *P. ferulacea* essential oil at different concentrations (1–200 µg/mL) [49]. After overnight incubation at 37 °C, MIC_100_ values were determined as the lowest concentration responsible for the absence of visible bacterial growth. Each experiment was conducted in triplicate and the reported result was an average of three independent experiments.

### 3.7. ABTS Scavenging Capacity Assay

The experiments, according to the reported method [50] with some modifications, are based on ABTS radical cation scavenging. The ABTS solution was prepared to achieve a final absorbance of 0.72 (±0.2) at 734 nm. Then, 1 mL ABTS solution was added to 100 µL of oil (1; 10; 100; 200; 250; 500 and 1000 µg/mL concentrations). Absorbance of ABTS was recorded after 6 min of incubation in the dark, at 734 nm. Finally, the absorbance was measured at 734 nm against a blank, and the percentage inhibition of ABTS radicals was determined from the following equation: ABTS•+ radical scavenging activity (%) = (1 − AS/AC) × 100, where AC represented the absorbance of the ABTS solution and AS was the absorbance of the sample at 734 nm. The concentration required for 50% inhibition was calculated as IC_50_. Each experiment was carried out in triplicate and the reported result was an average of three independent experiments. 

### 3.8. Hydrogen Peroxide Scavenging Assay 

The quantitative determination of H_2_O_2_ scavenging activity was recorded by the loss of absorbance at 240 nm, as described by Beers and Sizer [51,52]. Various concentrations of essential oil (1; 10; 100; 200; 250; 500 and 1000 µg/mL) were incubated in 1 mL of hydrogen peroxide solution (50 mM potassium phosphate buffer, pH 7.0; 0.036% (*w/w*) H_2_O_2_). After 30 min, the H_2_O_2_ concentration was measured at 240 nm. The percentage of peroxide removed was calculated using the following equation: peroxide removed (%) = (1 − AS/AC) × 100, where AC is the absorbance of 1 mL of H_2_O_2_ solution and AS is the absorbance of the sample at 240 nm.

### 3.9. Antioxidant Enzymes Measured in PMN Cells

The enzymatic antioxidant activity of superoxide dismutase (SOD), catalase (CAT) and glutathione *S*-transferase (GST) measured in PMN cells was determined using commercial kit protocols (BioAssay System, San Diego, CA, USA). The activity of the enzymes was expressed as U/L [53]. The essential oils of *P. ferulacea* were tested at a concentration of 25, 50, 100 µg/mL. The experiments were performed in the presence and absence of OZ (0.5 mg mL^−1^). 

### 3.10. Reactive Oxygen Species ROS Generation

The dichlorofluorescein (DCF) assay was performed to quantify ROS generation, following the protocol of Manna et al. [54]. The PMNs were treated with *P. ferulacea* essential oils at a concentration of 25, 50, 100 µg/mL with or without OZ (0.5 mg/mL), following the protocol of Napolitano et al. [55].

### 3.11. Statistical Analysis 

The data were examined by one-way analysis of variance (ANOVA), followed by Tukey’s multiple comparison post-hoc test. In Figure 2, Figure 3 and Figure 4, values are presented as mean ± st. err; numbers not accompanied by the same letter are significantly different at a *p* value < 0.05.

## 4. Conclusions

Plants of the *Prangos* genus are considered a great source of phytochemicals with therapeutic and economic applications. Given the growing demand for natural products, many *Prangos* species have been grown for their use in traditional medicine, the food market, the cosmetic industry and for ornamental purposes. In this work, the chemical and biological properties of the essential oils obtained from the leaves and flowers of *P. ferulacea* have been analyzed. From the GC-MS analysis, it can be observed that the two essential oils are characterized by the important presence of hydrocarbon monoterpenes, dominated by the high percentage of (*Z*)-*β*-ocimene. Although *P. ferulacea* essential oils showed only moderate antimicrobial activity, both samples exhibited good antioxidant activity in vitro and in particular, they caused a decrease in ROS and an increase in the activity of superoxide dismutase (SOD), catalase (CAT) and glutathione *S*-transferase (GST) in OZ-stimulated PMNs. Therefore, these essential oils could be considered as promising candidates for pharmaceutical and nutraceutical preparations.

## Figures and Tables

**Figure 1 molecules-27-07430-f001:**
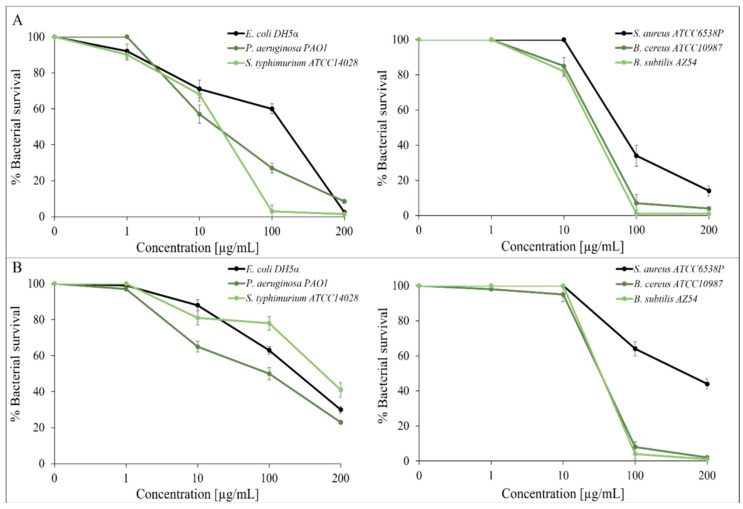
Antimicrobial activity of *P. ferulacea* flowers oil (**A**) and leaves oil (**B**) at different concentrations (0; 1; 10; 100 and 200 µg/mL) valuated by colony count assay, after 4 h of incubation, against Gram-negative E. coli DH5α, *P. aeruginosa* PAO1, *S. typhimurium* ATCC14028 and Gram-positive *S. aureus* ATCC6538P, *B. cereus* ATCC10987 and *B. subtilis* AZ54. The % of bacterial survival is represented by the y axis. The assays were performed in three independent experiments.

**Figure 2 molecules-27-07430-f002:**
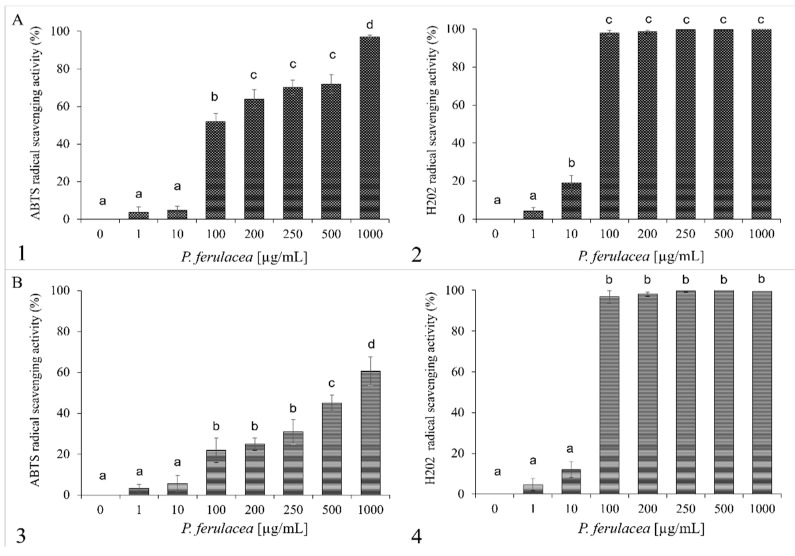
Antioxidant activity of *P. ferulacea* flower oil (**A**) and leaf oil (**B**). ABTS radical scavenging activity (1, 3) was measured after 10 min of incubation and reported as % of ABTS removed, with respect to the control. Hydrogen peroxide scavenging activity (2, 4) was measured after 30 min of incubation and reported as % of H_2_O_2_ removed, with respect to the control. Data were presented as mean and standard error and they were analyzed with a paired *t*-test. Bars not accompanied by the same letter were significantly different at *p* < 0.05.

**Figure 3 molecules-27-07430-f003:**
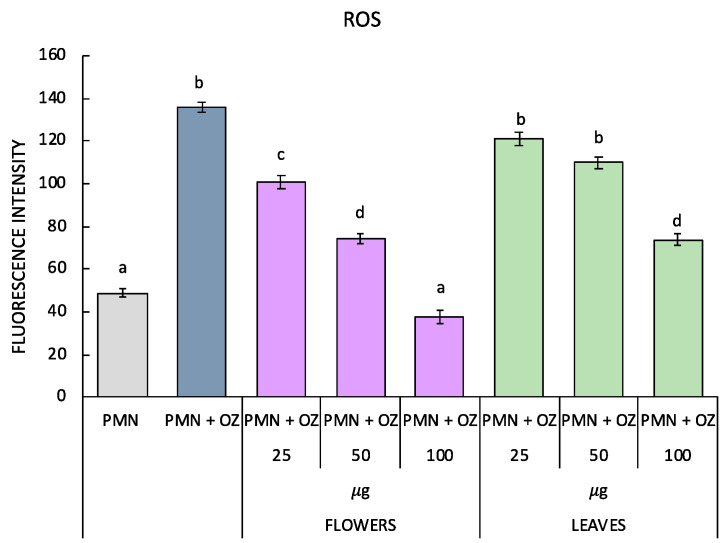
ROS production in polymorphonuclear cells treated with essential oils of *P. ferulacea* at concentrations of 0, 25, 50, 100 µg/mL with or without OZ (0.5 mg mL^−1^). Data were presented as mean and standard error and they were analyzed with a paired *t*-test. Bars not accompanied by the same letter were significantly different at *p* < 0.05.

**Figure 4 molecules-27-07430-f004:**
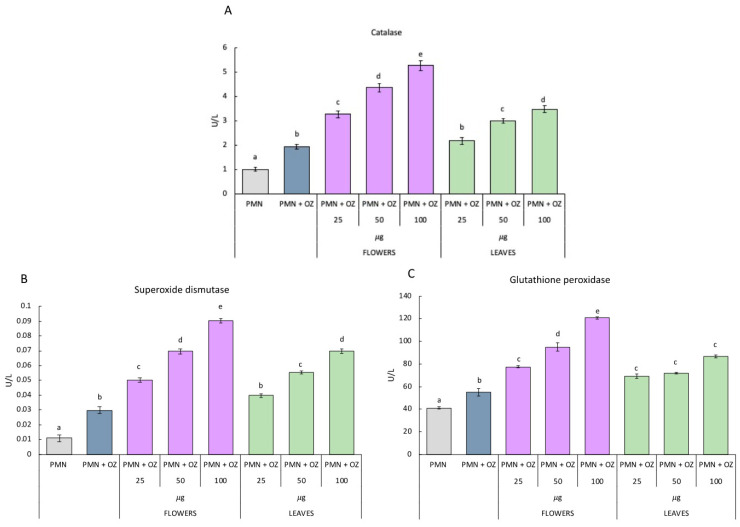
Activities of antioxidant enzymes: catalase (**A**); superoxide dismutase (**B**) and glutathione peroxidase (**C**) in polymorphonuclear cells treated with essential oils of *P. ferulacea* at the concentrations of 0, 25, 50, 100 µg/mL with or without OZ (0.5 mg mL^−1^). Data were presented as mean and standard error and they were analyzed with a paired *t*-test. Bars not accompanied by the same letter were significantly different at *p* < 0.05.

**Table 1 molecules-27-07430-t001:** Essential oil composition (%) of the flowers and leaves of *Prangos ferulacea*.

No.	Compounds ^a^	LRI ^b^	LRI ^c^	Flowers ^d^	Leaves ^d^
1	*α*-Pinene	1002	1017	4.28	2.26
2	3-Thujene	1024	1030	5.79	0.70
3	Camphene	1032	1037	0.21	0.13
4	*β*-Pinene	1087	1099	0.42	0.09
5	Sabinene	1111	1115	20.18	10.11
6	3-Carene	1149	1158	0.99	0.87
7	*α*-Phellandrene	1170	1174	0.66	0.09
8	*β-*Myrcene	1172	1176	1.64	0.87
9	*α*-Terpinene	1176	1179	3.03	0.33
10	Limonene	1187	1193	3.32	1.51
11	Sylvestrene	1201	1205	1.17	0.14
12	*γ*-Terpinene	1233	1238	8.09	-
13	(*E*)*-β-*Ocimene	1239	1243	-	4.62
14	(*Z*)*-β-*Ocimene	1244	1246	44.39	64.91
15	*α*-Bourbonene	1525	1528	-	0.13
16	4-Terpineol	1605	1611	2.08	-
17	Caryophyllene	1607	1612	-	7.65
18	*α*-Amorphene	1669	1675	-	0.72
19	Humulene	1682	1687	-	0.20
20	*γ*-Muurolene	1699	1704	-	0.31
21	Germacrene D	1703	1706	-	0.58
22	Cedrene	1807	-	0.28	-
	Monoterpene Hydrocarbons		-	94.17	86.63
	Oxygenated Monoterpenes		-	2.08	-
	Sesquiterpene Hydrocarbons		-	0.28	9.59
	Total		-	96.53	96.22

^a^ Components listed in order of elution on a DB-Wax column; ^b^ linear retention indices on a DB-Wax polar column; ^c^ linear retention indices based on the literature (https://webbook.nist.gov/ accessed on 3 October 2022); ^d^ percentage amounts of the separated compounds calculated from the integration of the peaks.

**Table 2 molecules-27-07430-t002:** Minimum inhibitory concentration values (MIC µg/mL) of *P. ferulacea* oils against Gram-positive and Gram-negative bacteria. Values were obtained from a minimum of three independent experiments.

Strains	*P. ferulacea* Flowers (µg/mL)	*P. ferulacea* Leaves (µg/mL)
*E. coli* DH5α	200	>200
*P. aeruginosa* PAO1	>200	>200
*S. tiphymurium* ATCC14028	100	>200
*S. aureus* ATCC6538P	>200	>200
*B. cereus* ATCC10987	100	100
*B. subtilis* AZ54	100	100

**Table 3 molecules-27-07430-t003:** IC_50_: concentration that inhibited 50% of the free radicals; ABTS: 2,20-azino-bis (3-ethyl-benzothiazoline-6-sulfonic acid); H_2_O_2_: hydrogen peroxide. Positive control was represented by ascorbic acid for ABTS; and resveratrol for H_2_O_2_.

Sample	IC_50_ of H_2_O_2_ (µg/mL)	Sample	IC_50_ of ABTS (µg/mL)
*P. ferulacea* flowers	60	*P. ferulacea* flowers	100
*P. ferulacea* leaves	50	*P. ferulacea* leaves	500
Resveratrol	0.05	Ascorbic acid	0.03

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
