# Peer review of "Chemical Composition and Biological Activities of Prangos ferulacea Essential Oils"

_molecules, 2022, doi:10.3390/molecules27217430_

Round 1

Reviewer 1 Report

In the current paper, the chemical composition, antioxidant activity and antibacterial potential of leaves and flowers of Prangos ferulacea grown in Italy were studied. The paper is well written and the results are properly discussed. In my opinion, the article can be published in Molecules after minor revisions as follows:

L51-53: write the scientific names in italic form. (check all text).

L100-115: The results of the analysis are slightly different from the previous studies, which, in addition to the origin, are also effective factors such as the harvest time, which should be mentioned.

L138: What is the reason for the sensitivity of Gram-positive bacteria to essential oils compared to Gram-negative bacteria? Refer to the following reference:

Hojjati, M and Barzegar, H. 2017. Chemical composition and biological activities of lemon (Citrus limon) leaf essential oil. Nutrition and Food Sciences Research, 4(4): 15-24

Figure 2. Put the lower letter on the columns to show the differences between treatments.

Table 3. What component did you use as a positive control? Write its name in the table.

The amount of compounds identified in flower and leaf essential oils were different.Could it be the main reason for the antioxidant activity of flowers compared to leaves?

 L249: How much water was added to dry matter of sample in the hydrodistillation?

 L252: The yield of oil extraction from flowers and leaves should be mentioned respectively.

L272: Where did the microorganisms prepare from?

L276:  Check the writing of “concentration. Samples”.

L331: Add the statistical analysis section after materials and methods.

Author Response

Reviewer 1

In the current paper, the chemical composition, antioxidant activity and antibacterial potential of leaves and flowers of Prangos ferulacea grown in Italy were studied. The paper is well written and the results are properly discussed. In my opinion, the article can be published in Molecules after minor revisions as follows:

1) L51-53: write the scientific names in italic form. (check all text).

A: Thanks. Done.

2) L100-115: The results of the analysis are slightly different from the previous studies, which, in addition to the origin, are also effective factors such as the harvest time, which should be mentioned.

A: A sentence was added.

3) L138: What is the reason for the sensitivity of Gram-positive bacteria to essential oils compared to Gram-negative bacteria? Refer to the following reference: Hojjati, M and Barzegar, H. 2017. Chemical composition and biological activities of lemon (Citrus limon) leaf essential oil. Nutrition and Food Sciences Research, 4(4): 15-24.

A: Therefore, essential oils rich in terpenes have been shown to possess good antibacterial activity. However, the weak activity against Gram-negative microbes could be due to the presence of their double membrane that protects the Gram-negative bacteria from the effect of the oil components. (Hojjati, M and Barzegar, H. 2017).

4) Figure 2. Put the lower letter on the columns to show the differences between treatments.

A: We used numbers to discriminate the different treatments.

5) Table 3. What component did you use as a positive control? Write its name in the table.

A: Done

6) The amount of compounds identified in flower and leaf essential oils were different. Could it be the main reason for the antioxidant activity of flowers compared to leaves?

A: It is assumed that the difference in antioxidant capacity between the two organs of the plant is due precisely to the different compositions of essential oils. The antioxidant efficacy of γ-terpinene, camphene, β-ocimene present in some essential oils has already been demonstrated by Negi et al. 2012. α-pinene and limonene may also be responsible for the antioxidant potential (Anamika et al., 2018). This would explain why a greater activity was found in the extracts of essential oils of flowers compared to those of leaves. In fact, the compounds mentioned are present in greater quantities in the essential oils of the flowers compared to the essential oils of the leaves with the sole exception of β-ocimene which, however, is also present in good quantities in the essential oils of the flowers.

7) L249: How much water was added to dry matter of sample in the hydrodistillation?

A: This detail was added in section 3.2.

8) L252: The yield of oil extraction from flowers and leaves should be mentioned respectively.

A: The yield was reported in section 3.2.

9) L272: Where did the microorganisms prepare from?

A: The microorganisms used are commercially available. These strains were grown in LB medium, under agitation at 37 °C.

10) L276:  Check the writing of “concentration. Samples”.

A: Done

11) L331: Add the statistical analysis section after materials and methods.

A: Done

Reviewer 2 Report

This article reported the various essential oils (EOs) and their bioactivities. This article needs several modifications before acceptance. My comments are-

  1. In the abstract, the Author should include how they identified these compounds.
  2. In the introduction, the Author should mention the importance of Eos and their uses.
  3. Lines 76, 106, 107, and elsewhere, “P. ferulacea” should be italic
  4. Lines 122-124, microorganism names should be in italics.
  5. Lines 143 and 148, should be in italics.
  6. In the whole study, the Author needs to clarify how EOs show antioxidant and antimicrobial activities. What are their mechanisms? What is the relationship between these compounds and their biological activities? Which compounds play the main role since β-Ocimene is abundant in the solution? More importantly, how can it be concluded that these activities come from EOs, not from other compounds?

Author Response

Reviewer 2

This article reported the various essential oils (EOs) and their bioactivities. This article needs several modifications before acceptance. My comments are:

1) In the abstract, the Author should include how they identified these compounds.

A: A new sentence was added in the abstract.

2) In the introduction, the Author should mention the importance of EOs and their uses.

A: A sentence was added in the introduction.

3) Lines 76, 106, 107, and elsewhere, “P. ferulacea” should be italic.

A: Corrected.

4) Lines 122-124, microorganism names should be in italics.

A: Corrected.

5) Lines 143 and 148, should be in italics.

A: The sentence was modified.

6) In the whole study, the Author needs to clarify how EOs show antioxidant and antimicrobial activities. What are their mechanisms? What is the relationship between these compounds and their biological activities? Which compounds play the main role since β-Ocimene is abundant in the solution? More importantly, how can it be concluded that these activities come from EOs, not from other compounds?

A: As regards the antimicrobial part, in order to determine or exclude a possible target towards which the P. ferulacea flowers and leaves oil is directed, we performed fluorescence microscopy experiments against Gram-negative bacterium E. coli as shown in figure S1. Bacterial cells used as control appeared intact and dark gray in optical, phase contrast, microscopy (panels A), developing blue fluorescence in the panels B. Bacterial cells treated with P. trifida essential oil from flowers (panel C) and from leaves (panel E) appeared not altered in shape and color, as shown in optical, phase contrast, microscopy. The same samples developed a blue fluorescence signal due to DAPI stain comparable to the control as shown in the panels D and F. These bacterial cells do not show any membrane damage, not developing red fluorescence due to IP dye entry. Many essential oils as the main mechanism of action are the alteration of the bacterial membrane permeability, but further mechanisms underlying the antimicrobial activity are also possible. Indeed, Wang et al. demonstrate, in their study, that Ginger essential oil is able to inhibit the energetic expression of some genes related to bacterial metabolism, to the cycle of tricarboxylic acid, proteins bound to the cell membrane and DNA metabolism (Wang et al. 2020). Therefore, essential oils rich in terpenes have been shown to possess good antibacterial activity. However, the weak activity against Gram-negative microbes could be due to the presence of their double membrane that protects the Gram-negative bacteria from the effect of the oil components (Hojjati, M and Barzegar, H. 2017). As regards the antioxidant part: It is assumed that the difference in antioxidant capacity between the two organs of the plant is due precisely to the different compositions of essential oils. The antioxidant efficacy of γ-terpinene, camphene, β-ocimene present in some essential oils has already been demonstrated by Negi et al. 2012. α-pinene and limonene may also be responsible for the antioxidant potential (Anamika et al., 2018). This would explain why a greater activity was found in the extracts of essential oils of flowers compared to those of leaves. In fact, the compounds mentioned are present in greater quantities in the essential oils of the flowers compared to the essential oils of the leaves with the sole exception of β-ocimene which, however, is also present in good quantities in the essential oils of the flowers.

Round 2

Reviewer 2 Report

I think the authors have now revised the manuscript, and it's ready to go for the next step.